# Formal Modeling of Responsive Traffic Signaling System Using Graph Theory and VDM-SL

**Afifa Nawaz** [1,*] , **Nazir Ahmad Zafar** [1,*] **and Eman H. Alkhammash** [2]

1    Department of Computer Science, COMSATS University Islamabad, Sahiwal Campus,
    Sahiwal 57000, Pakistan
2    Department of Computer Science, College of Computers and Information Technology, Taif University,
    P.O. Box 11099, Taif 21944, Saudi Arabia; Eman.kms@tu.edu.sa
*    Correspondence: afifajoyia555@gmail.com (A.N.); nazafar@gmail.com (N.A.Z.)

**Abstract:** Internet of things (IoT) is playing a major role in smart cities to make a digital environment. Traffic congestion is a serious road issue because of an increasing number of vehicles in urban areas. Some crucial traffic problems include accidents and traffic jams that cause waste of fuel, health diseases, and a waste of time. Present traffic signaling systems are not efficient in resolving congestion problems because of the lack of traffic signals. Nowadays, traffic signaling systems are modeled with fixed time intervals in which no proper mechanism for emergency vehicles is available. Such traffic mechanisms failed to deal with traffic problems effectively. The major objective is to establish a robust traffic monitoring and signaling system that improves signal efficiency by providing a responsive scheme; appropriate routes; a mechanism for emergency vehicles and pedestrians in real-time using Vienna Development Method Specification Language (VDM-SL) formal method and graph theory. A formal model is constructed by considering objects, such as wireless sensors and cameras that are used for collecting information. Graph theory is used to represent the network and find appropriate routes. Unified Modeling Language is used to design the system requirements. The graph-based framework is converted into a formal model by using VDM-SL. The model has been validated and analyzed using many facilities available in the VDM-SL toolbox.

**Keywords:** responsive time; traffic signals; shortest paths; VDM-SL toolbox; formal methods

## 1. Introduction

Traffic congestion has become a serious issue in large cities in terms of social, economic and environment impact. Smart cities aim to continuously improve the lives of citizens in different aspects: transportation, health, education, energy, etc. As cities now become larger, congestion is increased due to rapid urbanization. The smart city is a combination of multiple components of a specific area that is automated to make a digital environment and all smart devices and tools relate to a network. Traffic congestion has become a major problem because of the increasing population in huge cities around the world [1]. Some crucial traffic problems include accidents and traffic jams that causes a waste of fuel, motor power, property damage, environmental pollution, health diseases, and a waste of time. Every day almost 3700 people are injured due to traffic signal-related accidents, and 1.35 million people die every year because of traffic rule violations, careless driving, and technical faults [2].

Existing traffic signalized intersections exhibit several problems, mostly due to fixed signal time intervals. No proper framework for emergency vehicles, such as pedestrians, car drivers, police, accidents, fire brigade, and rescue services, is currently available [1]. Thus, these existing traffic signal systems failed to address traffic congestion problems efficiently. Due to the fixed clock cycle of red, green, and amber lights, vehicles need to wait at each traffic lane until the cycle is completed. In some cases, some vehicles must wait unnecessarily for the signal time of other lanes even if no traffic is occurring at the

signals [3]. Most vehicle drivers are aware of the traffic rules, but the system does not provide any support for a safe crossing signal, thereby resulting in road accidents.

Traffic signaling systems are one of the most important components of our daily routine.

There are severe problems with traffic congestion in urban areas due to lack of less efficient traffic signaling mechanisms. There is little work around traffic signaling systems in Formal methods and VDM-SL. Most traffic signaling systems are modeled with fixed time intervals in which no proper mechanism of emergency vehicles and safe crossing of pedestrians is provided at the signals. In this study, we develop traffic signaling and monitoring models with the integration of different approaches including UML, Graph theory, WSAN's, and VDM-SL formal method.

The objectives of our paper are presented as follows:

I.   Reduce the total waiting time at each signal intersection and develop a safe strategy to enable pedestrians and emergency vehicles to perform appropriate actions.
II.  Optimize efficiency by applying a traffic responsive strategy and increase signal dependency among signalized intersections to reduce queues, fuel, and power consumption.
III. Develop optimal paths from source to destinations using Wireless Sensor and Actor Networks (WSANs) and graph theory.
IV.  Validate the formal system using several facilities provided in the Vienna Development Method Specification Language (VDM-SL) toolbox to verify the correctness of the model.

Several possible solutions have been developed for traffic controls. However, improvements are still needed to overcome the issues in achieving efficiency at signals addressing responsive traffic monitoring and signaling system. In [4], public transport system estimation was presented based on license plate recognition (LPR) and cellphone location (CL) data. The use of several data sources, such as LPR and CL data, completely keeps the advantage of extreme level reporting and accuracy. Some disadvantages of CL data include the difficulty in estimating traffic flows. CL data keep focusing only on traffic state analysis, demand forecasting, and traffic zone. In [5], extraordinary resolution driving performance was utilized, and data are obtained through sensors from 303 drivers to monitor a driver's performance at the path section at joint level. This study needs some improvements, such as the use of traffic controllers to communicate with drivers, pedestrians, and emergency vehicles to construct an efficient system.

In this paper, we have modeled a Responsive Traffic monitoring and Signaling system using Unified Modeling Language (UML), Graph theory, WSAN, and VDM-SL formal method. Traffic communication and information collection play vital roles in transportation infrastructure. Unfortunately, most of the systems can only detect vehicle positions in a fixed manner, and their information collection task is only performed by communication and power supply cables, thereby leading to high construction costs. In another way, information collection and vehicle detection can be easily performed using WSAN because it provides the advantages of wireless distribution, flexibility without cables, and low energy consumption. The use of a wireless sensors network can solve many difficulties in traffic information collection systems [5]. Traffic signal strategies are classified as fixed cycle and traffic responsive. For the responsive scheme, the traffic signaling systems can be improved to set the dynamic time interval of red, green, and amber lights according to the amount of waiting vehicles and pedestrians at the signalized intersection to improve efficiency. A traffic signal is assumed to depend on the next signal at an intersection to reduce time and fuel. When an emergency vehicle is detected, the controller will alert each lane to stop the flow of vehicles and display the red signals on all lanes to continue the flow of the emergency vehicle. The traffic flow at a signal is measured to set a responsive traffic signaling scheme by analyzing the intersection. When a controller detects any unintended pedestrian near zebra crossing, all the signals will be turned to red to inform the vehicles to stop so that pedestrians can cross the road safely.

A VDM-SL formal model is developed by integrating approaches for a responsive traffic monitoring and signaling system that enhances the efficiency of vehicles and increases safety at the traffic signals. This model will help make schedules between the occurrences of traffic signals in the targeted areas to provide a successful flow of vehicles across all traffic signal intersections. Our graph-based model is converted into a formal model through the VDM-SL. The proof of validation is provided by various facilities that are available in the VDM-SL toolbox [6]. The formal specification is checked by type and syntax checkers, pretty printer, and integrity for validation of the system.

Figure 1 describes an abstract representation of a traffic signaling intersection. It displays different signalized intersections, cameras, traffic signals, emergency vehicles, primary vehicles, and other objects. The proposed specification is based on finding different locations of places, determining the shortest path toward the destination path in terms of distance and time, and identifying less traffic on the signalized intersection. For traffic jam or rush, dynamic information is provided to update the controller using signal intersection video cameras that take real-time video streaming and images of the traffic situation to find the shortest routes. The shortest route is estimated depending on two factors, namely, distance and time. For a real model, the shortest route has equal importance in terms of distance and time. The distance-dependent or time-dependent shortest route may be selected as required.

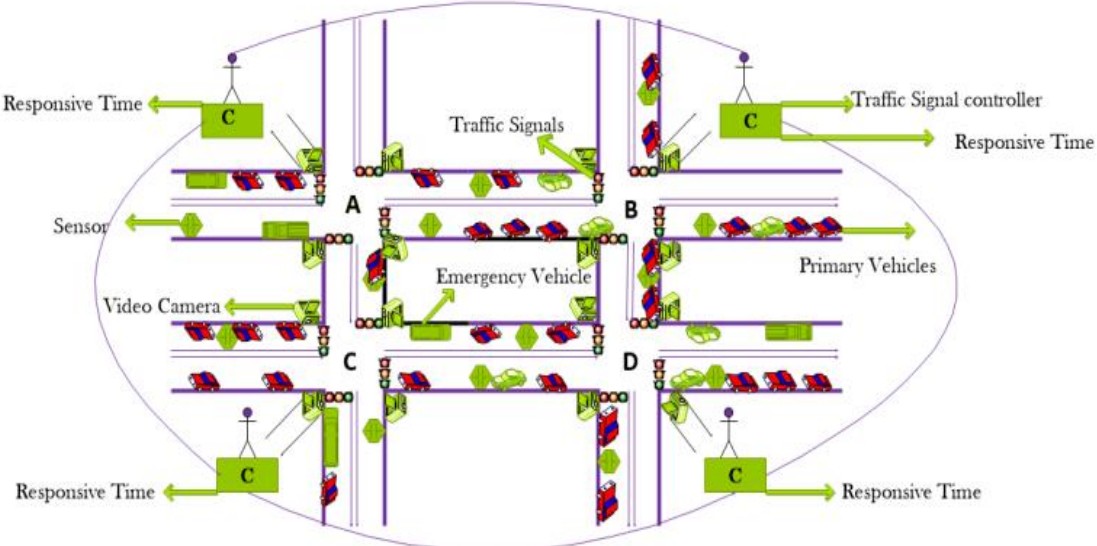

**Figure 1.** Abstract representation of Traffic Signalized Intersection.

The remainder of this paper is organized as follows. Section 2 discusses the Background. Section 3 presents our Contribution. Section 4 presents the Related work. Section 5 explains the System Architecture. Section 6 defines Introduction to Formal modelling. Section 7 explains the formal specification. Section 8 explains the Modeling Components and Properties of System. Section 9 describes the Formal Analysis with VDM-SL Toolbox Facilities. Section 10 draws the Conclusion. Section 11 draws the Future Work.

## 2. Background

UML is a standard used to visually define an object-oriented program and helps in planning, visualizing, and organizing a system. UML is mostly used for different purposes, and its reusability and reliability make it the best choice for the developers [7]. The UML will be used to design the system requirements and define functionality of the system.

Graph theory is used to represent the whole network and in building structural models. A graph-based topology is integrated with WSAN. It represented the whole network of traffic signals that act as nodes, and the connectivity of objects that are represented as edges

can build a network. Through this network, the optimal paths and required location can be found based on time and distance.

Formal methods are applied in managing and developing computer programs that are mathematically nature-based techniques for describing system properties [8]. Formal method provides a framework wherein we can develop, verify, and specify a system systematically. Formal methods are efficient techniques used to formalize valid and invalid data. Formal and semi-formal methods are two well-known techniques in writing formal specifications. Requirements can be easily converted from semi-formal to formal. Various kinds of formal languages, such as Z notation, B method, and VDM++, are used for formal modeling.

### 3. Our Contributions

Formal Modeling of Responsive Traffic Monitoring and Signaling System considering:

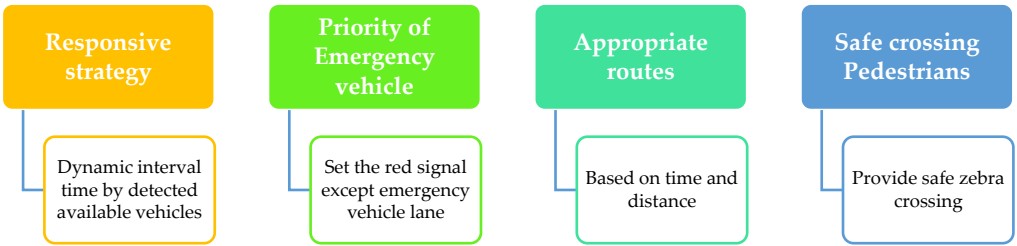

- In the proposed model, wireless sensors and cameras will be assumed for collecting the information and actors will be used for communication and decision making.
- Graph theory will be used for representing the network and finding appropriate routes.
- Unified Modeling Language (UML) will used to design the system for capturing the requirements.
- The graph-based framework will be converted into formal model by using Vienna Development Method Specification Language (VDM-SL).
- The evidence of validation will provide through many facilities included in VDM-SL toolbox.

### 4. Related Work

Much work is dedicated in this area. However, only the closely related one is presented here. In [1], the traffic congestion problem is addressed to enhance the number of vehicles that cross the traffic signal intersection by keeping the balance of road signals using Q Learning technique, which is a model-free reinforcement learning. Q-learning is a type of reinforcement knowledge that uses a trial-and-error technique to find the difficult and stochastic framework and the best attitude based on experiments. In [4], the author dealt with the transport infrastructure of urban areas, and the traffic flow estimation was based on mobile phone location and plate recognition based on license data. The proposed study uses two methods to filter phone location data and fetch the temporal traffic infrastructure features for a specific road intersection. The main goal is to estimate the traffic flow on a single road using filtered mobile location data, fetches, and extracted features of temporal traffic. In [5], a study is carried out to optimize the rising resolution of driving behavior and data gathered via smartphone sensors to analyze driver behavior at the signalized intersection and jam stage. Such sensor information is paired with the characteristics of traffic and road structure that are interpreted using the tools from the geographical information system. In [7], LPR information was detected by electronic police devices that became increasingly available at the urban traffic signalized intersection. It presents an idea to develop a set of innovative algorithms to determine the control parameters against traffic situations responsively. The Bayesian network model is used to construct an autonomous intelligent traffic light system to make decisions that are based on the complex Bayesian network, as in [9]. In [10], a technique was introduced to determine traffic volumes using

trajectory data from connected vehicles and navigation devices. The design of vehicle appearances at traffic signals based on the Poisson method is used to compensate for signal coordination. A technique is introduced as a set of innovative approaches based on the philosophy of probability in [11]. The proposed methods aimed to develop and solve a standard-variable equation for the penetration level of probe cars and trucks by leveraging the waiting locations of the vehicles throughout the lines.

In [12], an Apache Spark-based geo computing platform is designed to estimate the vehicle miles traveled using the derived broad GPS data. This platform has a technique called map matching module, which is used to match uncertainty. In [13], a co-simulation-based optimization approach that prioritizes trucks was used because all involved vehicles will benefit from this priority. This system is provided with a road network simulator to make consistent improvement in traffic delays, vehicle stops, and fuel consumption. In [14], the hybrid flow of autonomous vehicles and human-driven vehicles has numerous equilibrium relation flows. These measures are not only used to reduce congestion on the platform but also to shorten the distance among the roads as consumers select their route selfishly. In [15], conflict-based security and safety performance functions were developed using the generalized linear model (GLM) approach at signalized intersections. Some traffic variables, such as maximum queue length, traffic volume, platoon ratio, and shock wave attributes, are used to identify the real-time traffic situation. The video analysis procedure is recorded by collecting real end conflicts and by using traffic attributes at each signal from a recorded video. In [16], a novel-based approach was used to provide the dynamic traffic signal cycles and different signal phases of isolated intersections. The solution is dynamic lights control system that combines a wireless sensor network (WSN) for real-time monitoring with numerous fuzzy logic traffic controllers. Each rule for each phase depends on parallel roadways, using this technique.

In [17], a central server model that produces a great asset was developed to enhance the performance and reduce the machine cost through a web controller and microcontroller system by using a global positioning system and fuzzy controller system approach. This infrastructure controlled traffic dynamically and assigned the traffic lights automatically. In [18], the traffic congestion problem at the multi-vehicle signalized intersection and multi-road intersection area were addressed using image processing and beam interference strategies. It offers an integrated time management framework, in which time management is complex for the movement of every traffic lane, and the timetable is carried out in real-time. In [19], the author proposed a decentralized reinforcement learning method that incorporates a dynamic scheduling technique to solve traffic congestion problems. This approach relies on detailed real-time traffic analysis. In [20], the issue of controlling traffic signals under fixed interval time and shortest paths in urban areas networks was addressed. Backpressure strategies are used to tackle the network layer on corresponding paths. In [21], a solution was presented to overcome the traffic congestion issues and improve the overall progress in WSNs that efficiently works for the rush loaded or idle nodes. The author followed the study in dynamic interval time to deal with congestion problems using a potential-based traffic dynamic routing algorithm.

In [22], a study of WSN-based real-time traffic monitoring and simulation, network management, and identification of network anomalies is developed. With this technique, traffic flow information and its related average travel time are used for obtaining accurate measurements with an optimization algorithm. In [23], an intelligent traffic signal system based on the applications of WSN is presented. At road intersections, a mechanism for controlling vehicle length on roads during the red signal cycle is described to perform better control in the green signal cycle. In [24], a stochastic hybrid scheme was used to develop a stochastic flow model (SFM) at a single intersection that overcomes traffic congestion issues. The quasi-dynamic method depends on a partial state data set by detecting the vehicle in different situations and without counting the need for a specific vehicle. In [25], the crash causative parts of signalized crossings were investigated by following different traffic directions using advanced mathematical models. Logistic regression models and hi-

erarchical Poisson regression have been used to determine the severity of traffic signalized intersections and crash frequency approaches. In [26], pedestrian's safety was enhanced by providing the right way for pedestrians using marked crosswalks at signalized intersections. For this purpose, pedestrians' intersection behavior and perception data have been managed from 55 signalized intersections in Kolkata, India, and a numerical connection is generated within the part of pedestrian sign breach and the number of deadly pedestrian collisions. In [27], this research aimed to improve road safety using the Internet of Things. This system takes the workload of the road and manages the traffic light signals according to the number of vehicles. Traffic light logic is changing periodically, thereby possibly increasing the safety of the control system by analyzing the real-time information they received. In [28], the hybrid lighting technology was used to overcome the dim traffic signal issue that results in traffic rule violations. The designed technology has improved traffic signal sources for futuristic smart cities and car autonomous systems for retrieval and modeling, image recognition, and simulation.

In [29], multiple agent reinforcement learning methods are used to overcome traffic congestion issues by increasing the cooperation of traffic signals. By presenting a knowledge-based communication protocol, every single agent will access collective data representation for the traffic infrastructure collected by all other agents. In [30], the study aimed to determine the adaptive behavior of cyclists and pedestrians toward a green light. The current countdown timers caused cyclists to improve their speed to the required green sign, and controlled origin-up stops have been marked in the regulating circle because of its warnings of several red-light ruins. In [31], the application of a decentralized signal management scheme that considered the reliability of traffic mobility and its effect on the environment was suggested. A reinforcement learning algorithm was used to set the duration of each turning movement. In [32], a technique called deep reinforcement learning displayed its feasibility in making intelligent traffic signal controls. Cooperative multi-agent group-based reinforcement learning is a new framework that is introduced based on a cooperative car infrastructure system with k nearest neighbor to realize the efficient control in large area networks. In [33], multiple agent reinforcement learning methods are used to overcome traffic congestion issues by increasing the cooperation of traffic signals. By presenting a knowledge-based communication protocol, every single agent will access collective data representation for the traffic infrastructure collected by all other agents.

In [34], the dilemma lengths and zone types are investigated. Moreover, a link between two models and the probability of stopping modules have been discovered. Multiple linear regressions are used to declare the maximum distance passing boundaries and minimum stopping boundaries, and logistic regression based on binary type also increased the dilemma zone. In [35], a multi-agent system was presented based on a hierarchal structure that involved two levels of traffic lights. At every level, an agent used the reinforcement learning algorithm to train long short-term memory neural networks for real-time traffic prediction. In [36], swarm-based heuristic optimization algorithms are used to address real-time traffic situations. The proposed solutions and their performance have been verified using the SUMO simulator tool. In [37], the key problem is addressed by showing how to measure the arrival rate of the interlinked lanes of every phase for the traffic light plans of two adjacent intersections. The transition probability of one interlinked intersection and other intersections was utilized to measure the arrival rate of connection lanes.

In this study, we are constructing an efficient traffic signaling and monitoring system by integrating different approaches, including UML, graph theory, WSAN, and formal methods, to improve the efficiency among signal intersections. We work on different mechanisms, such as a responsive traffic signaling scheme, a mechanism for emergency vehicles, and the allocation of the appropriate paths and pedestrians crossing at signal intersections so that vehicles can travel smoothly in less time, thereby minimizing the number of vehicles stops, reducing accidents at traffic signal intersections, and saving fuel and time. Our system will monitor the real-time traffic situation and communication between

signal intersections to provide information about the present situations. It identifies the pedestrians and provides safe crossing and appropriate paths and allocates the minimum distance path to save time.

## 5. System Architecture

System architectures refer to different design decisions nature-wise; these designs are mostly commercial and nontechnical decisions. In any system, designing the model through UML diagrams is important to meet the functional and nonfunctional requirements of the system.

In this study, we are constructing an efficient traffic signaling and monitoring system with the integration of different approaches, including UML, graph theory, and VDM-SL formal methods, to improve the efficiency among signal intersections. Flow chart of the proposed techniques is presented as in Figure 2:

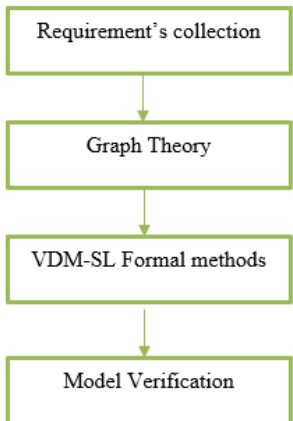

**Figure 2.** Flow chart of the proposed techniques.

### 5.1. UML Based Models

By using UML, we will define how the signals will periodically update according to the real-time conditions of the roads and how the signals depend on each other at the intersection. We modeled our system using various diagrams, such as use case and Sequences.

Different types of diagrams are designed, and the details are given below:

Use cases are a set of different types of scenarios that define the interaction between a system and a user. The two major components of a use case include actors and use cases.

Sequence diagrams model the behavior of systems by defining the way in which multiple objects interact with each other to achieve the goals. The sequence diagrams are read in ascending and descending order. In a traffic system, all the modules are required to work online. As such, any system can send requests at any time. An information system always has a strong behavior. Therefore, some behavior of this work is complete in the sequence direction. Thus, this kind of diagram is known as the sequence diagram.

Figure 3 displays the extended use case diagram and defines all the details of this system, which includes five actors. They are all detectors that are detecting the current situation on the traffic signals. Traffic controllers are continuously monitoring and taking decisions based on the current traffic flow. Traffic detectors detect all situations on the road and pass information to the controller. Storehouse stores the details about the detectors, vehicles, pedestrians, and emergency vehicles. Pedestrians and primary vehicle drivers will know the real-time road condition of signals and then decide whether to pass over the lane and/or wait for the other signals.

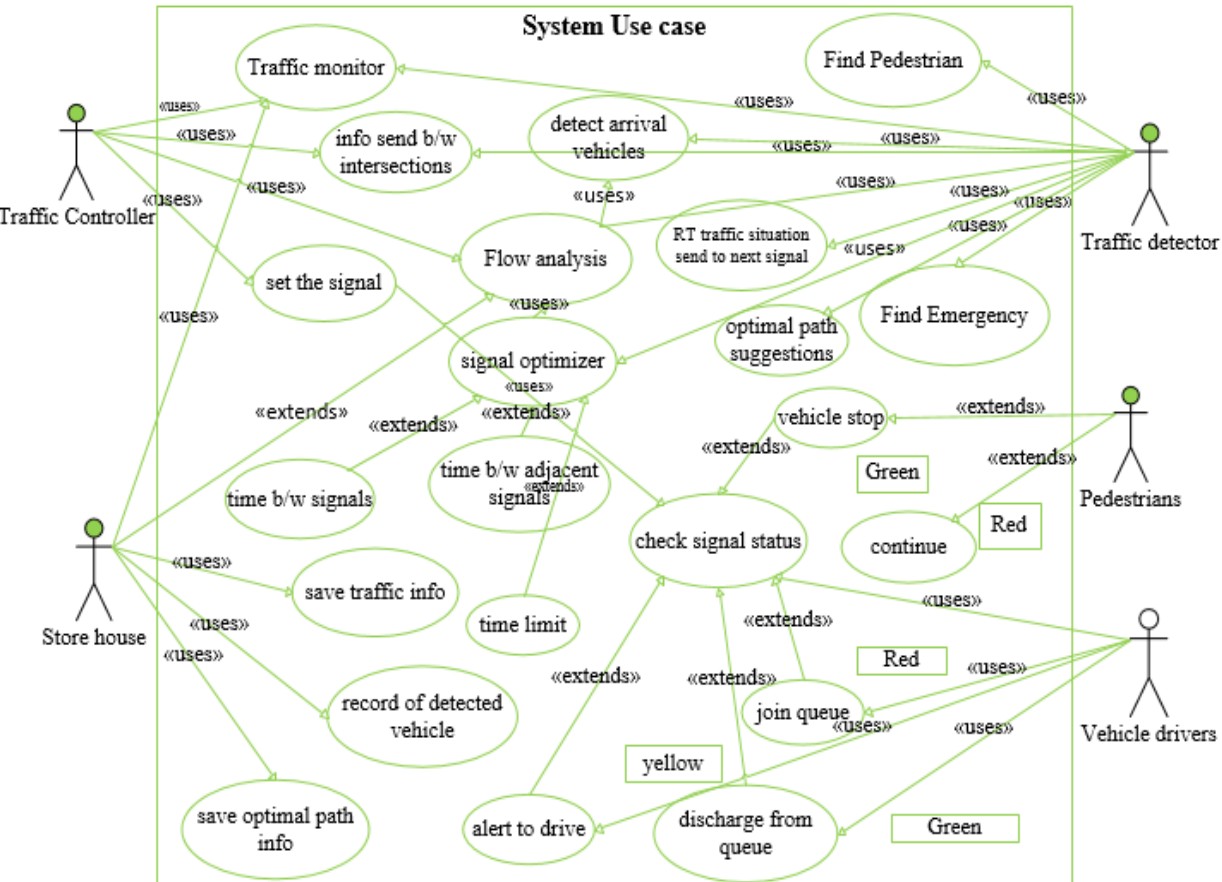

**Figure 3.** Extended use case.

Figure 4 shows an object of primary vehicles that crosses the detector. The detector will detect all vehicles and count the vehicles at every lane. Subsequently, the detector sends the current traffic situation to the traffic controller for future actions. Sensors are sending and receiving information at the parallel time. In this way, it identifies the current flow of traffic with the help of detector sensors.

The traffic controller collects all the information, analyzes this received information, and sends the traffic flow to the traffic optimizer. After completing all processes, the traffic controller sets the signal and responsive time at numerous vehicles' lane. Primary vehicles check the signal and follow the time to cross the road. According to dynamic time, two lanes can possibly have different times to cross the vehicles by counting the vehicles at every lane, and the remaining lane must be stopped.

Figure 5 shows when objects of primary and emergency vehicles cross the detector. The detector detects the emergency vehicle lane at this time and will send data to the controller for further actions. The controller collects all information from detectors and passes current traffic information to the optimizer, which optimizes and sends the received data back to the controller. The traffic controller sets the responsive time on the signalized intersection at a specific lane. Priority is awarded to emergency vehicles, such as fire brigades, police, and ambulance, by turning signals in nearly all vehicle lanes, except lanes where emergency vehicles are crossing, into red. Other primary vehicles must stop because the emergency vehicle may pass in any lanes.

Figure 6 shows when the pedestrians will reach the footpath; detectors will act as cameras and sensors that detect the pedestrians, send information to the traffic controller, detect all pedestrians, and count the pedestrians at every lane. Next, the detectors send the current traffic situation to the traffic optimizer for future actions.

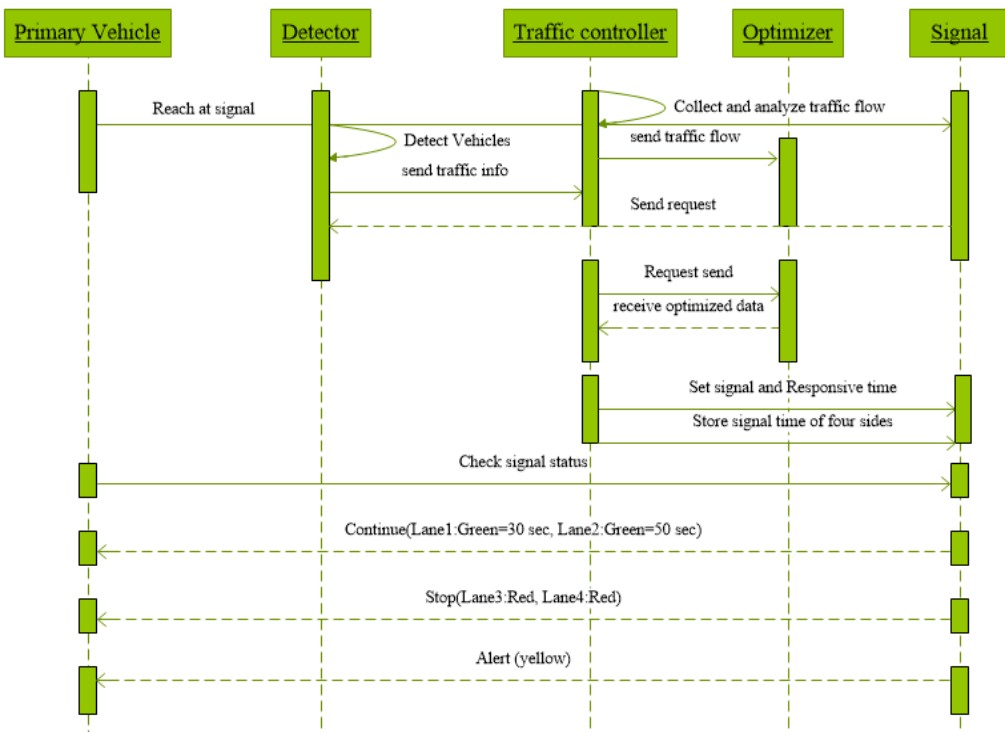

**Figure 4.** Sequence Diagram for Primary Vehicles.

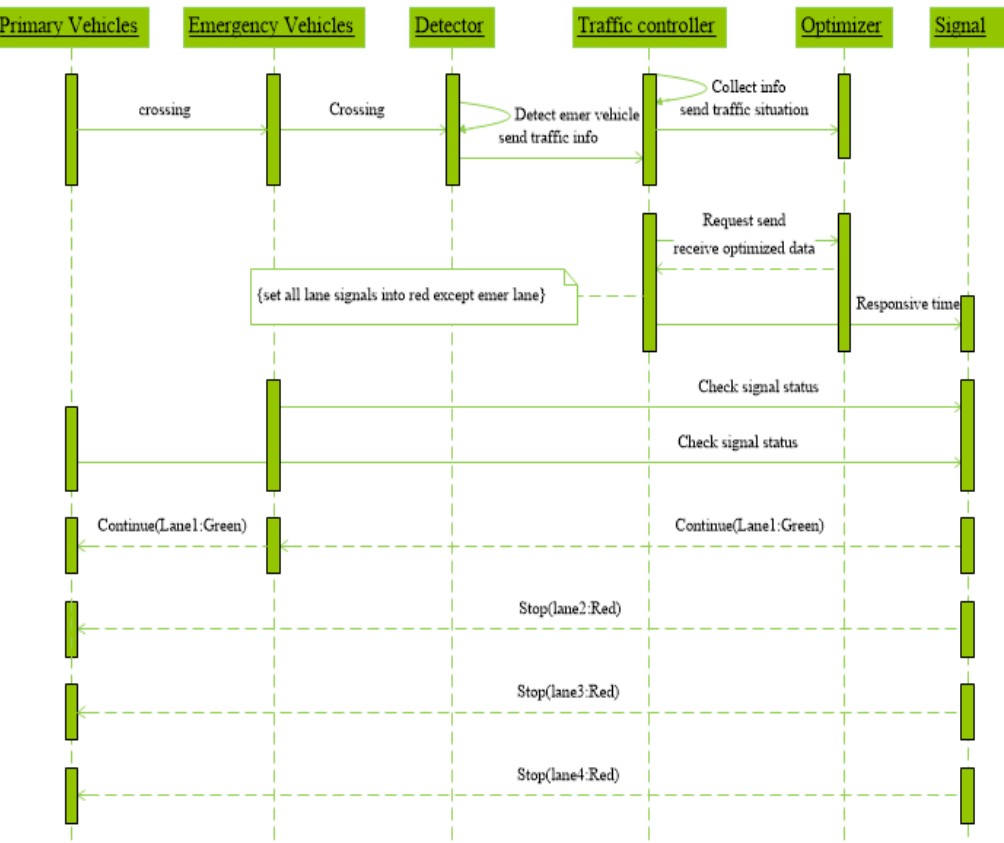

**Figure 5.** Sequence Diagram for Emergency Vehicle.

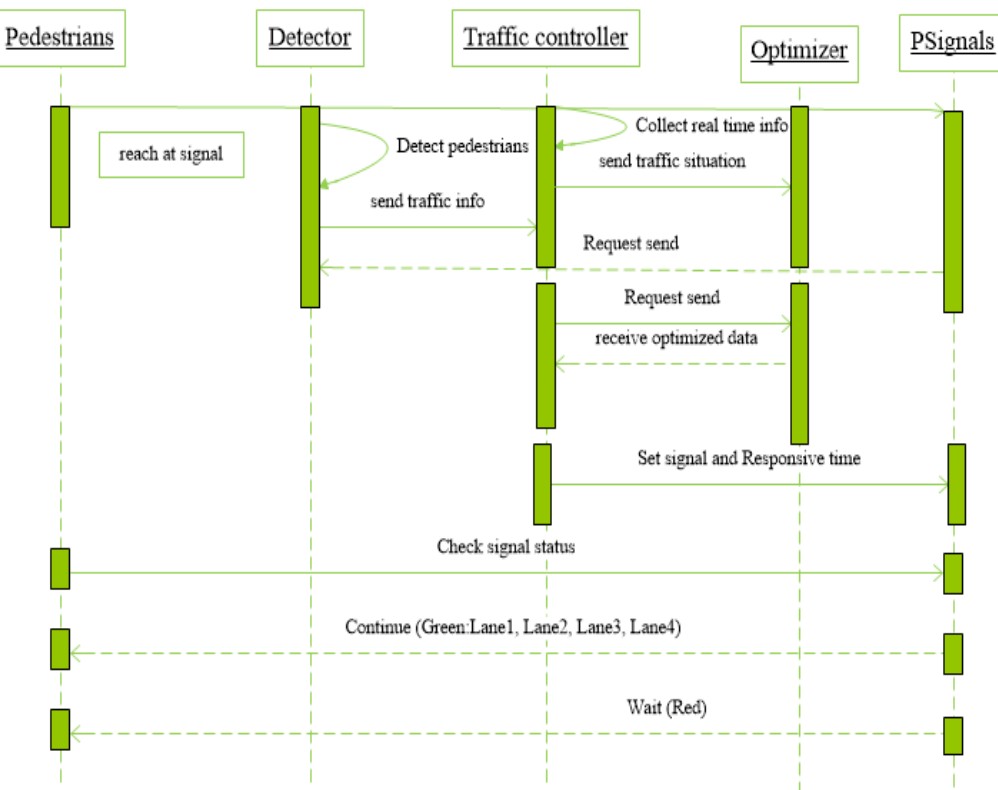

**Figure 6.** Sequence Diagram for Pedestrians.

To increase efficiency, detectors will send and receive information in real-time. The traffic controller collects all the information after analysis and sets the pedestrian signals according to responsive time. In this way, the efficiency of the system will be improved. Pedestrians can check the specific lane signal to cross the road by following the responsive time. When pedestrians are crossing, primary vehicles must wait until they finish.

*5.2. Graph Theory Model*

Initially, we present the real traffic environment in graph theory. Graph theory represents the entire traffic signal network that acts as nodes, and the connectivity of signals is represented as edges to build a network, and through this network, the appropriate routes have been found. For signal dependency, the speed efficiency of vehicles between traffic signal intersections must be improved with graph-based topology. As a result, every vehicle will reach its destination smoothly without wasting time, stops, and fuel consumption. In Figure 7, the speed efficiency of vehicles between traffic signal intersections is improved with graph-based topology so that every vehicle reaches its destination smoothly without wasting time, stops, and fuel consumption.

Information collection and vehicle detection can be easily performed using WSANs because of its advantages in wireless distribution, flexibility without cables, and low energy consumption. The use of wireless sensor networks can solve many difficulties in traffic information collection systems [10]. Using WSANs, the communication of one intersection to another intersection about road information will be provided with a group of sensors, cameras, and devices. WSANs work along with different nodes to perform multiple tasks for collecting information, monitoring traffic, and analyzing collected data at specific regions.

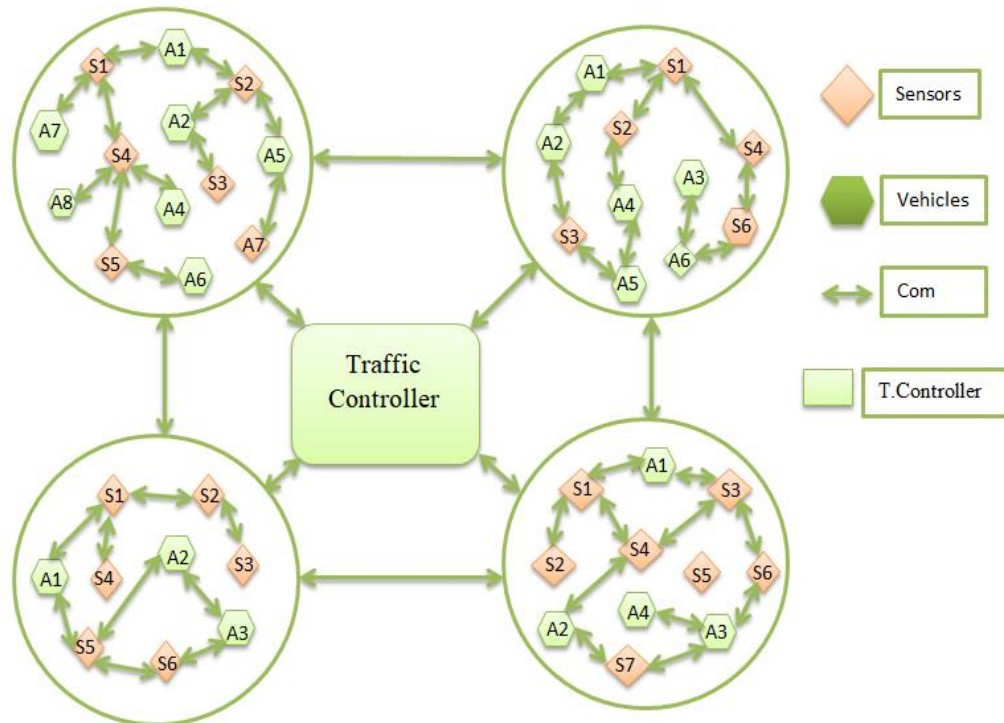

**Figure 7.** Graph-based Topology using WSANs.

## 6. Introduction to Formal Modelling

Formal methods are math-based languages, technologies, and tools that can be applied to any part of the program life cycle. Formal methods assist in the development of critical systems by providing an abstract and clear description of the function of the mechanism. Expressions used in formal methods are called formal specification languages. Formal regular languages are based on set theory and first-order predicate computation. The language has formal semantics that can be used to express a specification in a clear and unambiguous specification.

The formal specification of the developed model is defined using the VDM-SL language. In this work, different notations and symbols of the VDM-SL are used like composite objects with types, quote types, set, numeric data types and sequence. This step is having two parts that are static part and the dynamic part. In the static part, the composite objects are defined that are as important data types. A composite object entails several fields that have different types and invariants which limit their actions. In the dynamic part, the most important is state and operations are well-defined. In the operations, pre and post conditions are described for the accurate operation of the model.

In our proposed model, three different modules are included.

1. Responsive time
2. Emergency services
3. Optimal paths.

They have certain identifiers with token types, enumeration types, composite objects, states, functions, and operations. Through class diagram, enumeration types and record types in programming language are transformed as quote and composite type in the VDM-SL.

### 6.1. Responsive Time

In Figure 8, the number of waiting vehicles will be counted at each lane and then set a responsive traffic signaling scheme that based on analysis to cross the intersection.

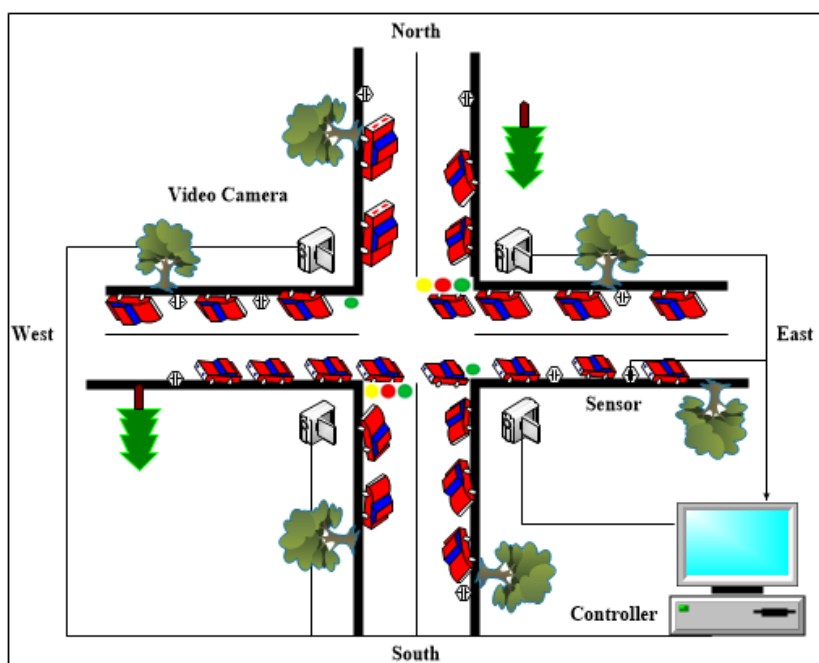

**Figure 8.** Responsive time for available vehicle at one path.

### 6.2. Emergency Services

In Figure 9, for the occurrence of emergency vehicles, a mechanism has provided that identifies the emergency vehicles lane through sensors and then send their information to the controller who has change all traffic signals into red except that emergency vehicle lane. Cameras is used to monitor the real-time conditions at each signal intersections to estimate the congestion, identify the emergency vehicle, and primary vehicles. If there is an emergency vehicle like ambulance, police cars and fire brigade detected by the sensor and camera at an intersection; sensors send real-time information to the controller. When the emergency vehicle will be detected, then the controller will alert each lane to stop the flow of their vehicles and display the red signals on all lanes to continue the emergency vehicle flow. The controller is providing the response time at the emergency vehicle occurrences lane. In this way, priority is provided by emergency vehicle so that they do not have to unnecessarily wait at the red signal.

Four traffic signal intersections are shown in Figure 10; our system provides real-time road information and information to the driver about heavy vehicles and congestion paths. Drivers can choose the shortest path to reach at the destination according to distance and time. When the controller detects any pedestrian on the footpath near zebra crossing then all the signals will be turned into red lights to inform all vehicles to stop, so that pedestrians can cross the road safely.

### 6.3. Optimal Path

In Figure 11, different signalized intersections, traffic lights, and objects are included. Our model is finding the different locations of places to obtain the shortest path toward the destination path in sense of distance and time and identify the less traffic on the signalized intersection. To provide the traffic jam or rush, dynamic information is provided that is updated into a controller updated by signal intersection video cameras that take video streaming and images of current time traffic situation to find the rush on routes. The shortest route is estimated that depends on two factors that are distance and time. For a real model, the shortest route has equal importance in a sense of distance and time. If a vehicle owner or any person is related to distance, he may use distance-dependent shortest route otherwise he may be used time-dependent shortest route is selected. Every vehicle driver can see two or more ways to reach at destination and find the suitable path to save

time and fuel consumption. For the signal dependency, one should improve the speed efficiency of vehicles between traffic signal intersections with graph-based topology so that every vehicle will reach at destination smoothly without wastage of time, vehicle stops, and fuel consumption.

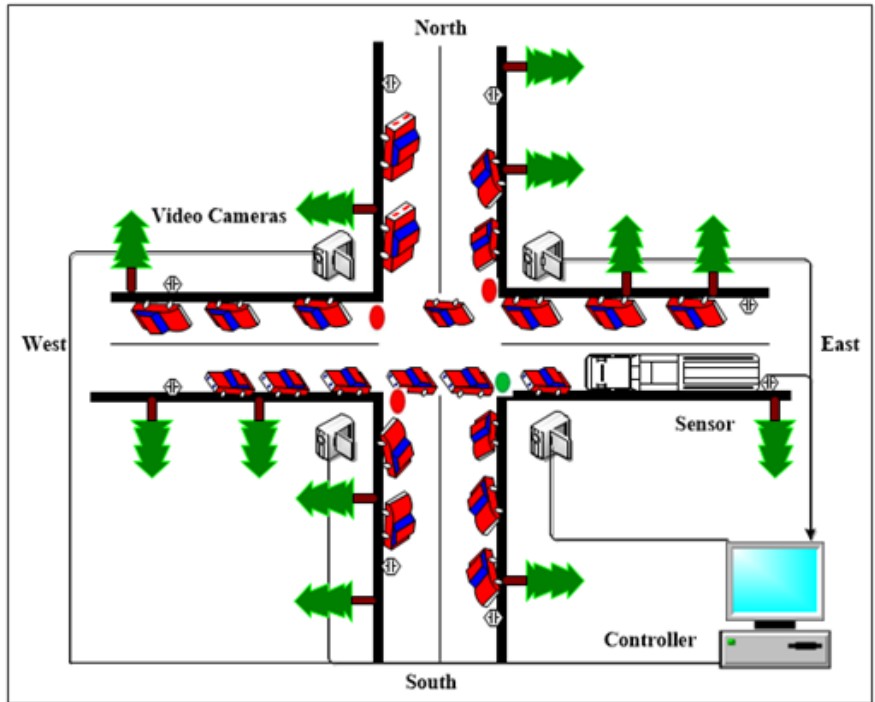

**Figure 9.** Emergency Vehicle Detection.

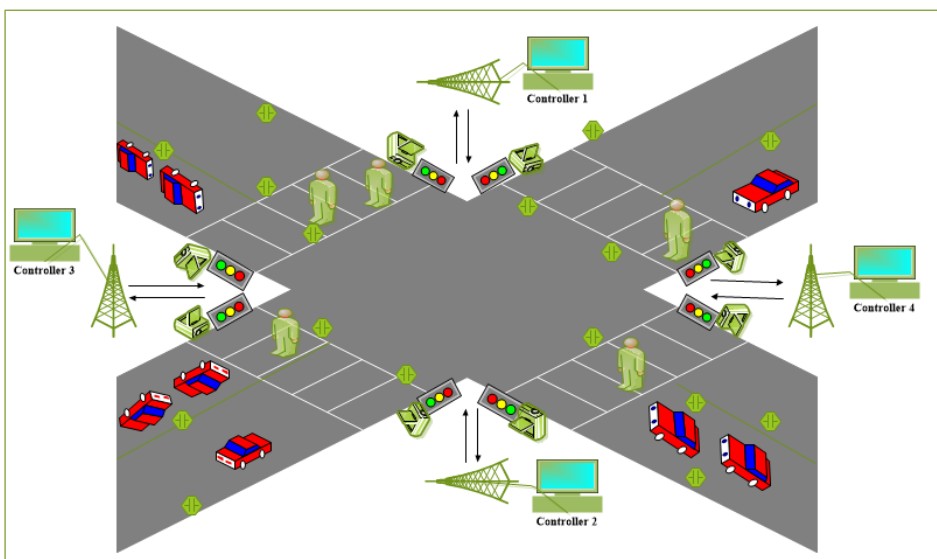

**Figure 10.** Pedestrians and primary vehicles detection.

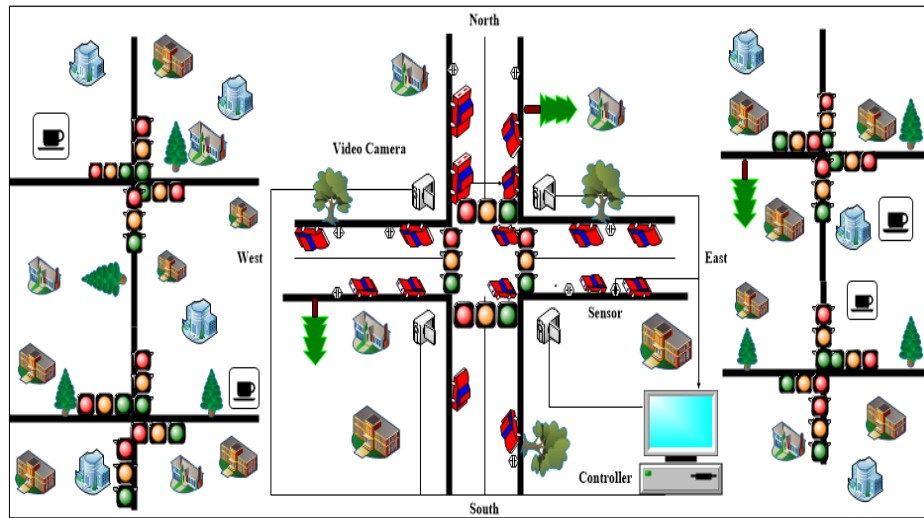

**Figure 11.** Optimal path.

## 7. Formal Specifications

UML is a software design specification language that includes software components, their behavior, and relationships. UML is a graphical representation of software design and collects the requirements of the system. To convert from UML to formal model, we can easily collect system requirements in an earlier phase [8]. In this research, we are presenting formal specifications for the basic components of our system using UML including use cases and sequence diagrams. The formal specification is based on basic mathematical notations like mapping, set, sequence, predicate logic, and relations. Our formal specifications of three modules are given below.

### 7.1. Responsive Time Module

In the formal specification of the traffic system, one part has types in which variables with their quote types are declared. This model is using various variables, such as traffic, vehicle type, quote-type traffic situations, and some are token types and string types.

```
types
string = seq of char;
TSignals = <Red>|<Yellow>|<Green>;
PSignals = <Red>|<Yellow>|<Green>;
EVehicles = <FireBirgade>|<Ambulance>|<Accident>|<Police>;
Traffictype = <EVehicles>|<Pedestrians>|<PVehicles>;
Time = nat; Id = token;
Name = token; limit = token;
route = token; RT1 = token;
RT2 = token;
trafficSitutions = <Stopped>|<Congestion>|<Smooth>;
Sensorinformation = token;
Location = String; CLocation = String;
Sensors = String; Cameras = String;
```

The most important composite objects are vehicles, emergency vehicles, and dynamic time and conflict. Dynamic Time object has an invariant for traffic time that has a limit of 24 h, 60 min, and 60 s. Conflict object has an invariant for opposite routes that are not equal to the same signals and equal time at the same time because this condition prevents the chances of accident. The controller object has signals and a set of conflicting paths and their signal timer. It has an invariant, that is, all conflicted paths have signals, and one of the conflicted paths is red while the other is not red.

```
Vehicles:: vid: Id
vloc: int
vtime: DTime
tsignals: TSignals;
Evehicles:: evid: Id
ename: Name
etime: DTime
esignals: ESignals;
Pedestrians:: pid: Id
ploc: int
ptime: DTime
psignal: PSignals;
DTime:: hour: nat
min: nat
sec: nat
inv mk_DTime (h, m, s) == h<24 and m < 60 and s < 60;
Conflict:: R1: RT1
R2: RT2
inv mk_Conflict (R1, R2) == R1 <> R2;
Controller:: Signals: map route to TSignals
conflicts: set of Conflict
timer: map route to DTime
inv mk_Controller (sls, cfs, tr) == forall c in set cfs & c.R1 in set dom sls and c.R2 in set dom sls
and c.R1 in set dom tr or c.R2 in set dom tr and (sls(c.R1) = <Red> or sls(c.R2) = <Red>);
```

In the second part of the formal specification values; we used predefined values that have the limit of primary vehicles on every road.

```
values
VehiclesLimit: int = 80;
PedestriansLimit: int = 40;
```

The important part of the formal specification is called state. The state explains the behavior of the system and assigns different values to variables. Different variables are defined with types in the state portion. The state has an invariant and initialization. Invariant is the condition that must be applied successfully before the declaration of objects and initialization in the state that defined the values of all variables. Then, the traffic system will check the primary vehicles that must be equal to or less than the fixed limit on one lane.

```
state TSystem of
location: map Sensors to Location
clocation: map Cameras to CLocation
currenttraffic: set of trafficSitutions
traffictype: set of Traffictype
pedestrians: set of Pedestrians
vehicles: set of Vehicles
evehicles: set of EVehicles
controllers: set of Controller
conflicts: set of Conflict
inv mk_TSystem (-, -, -, v, p, -, -, -, -) == card v <= VehiclesLimit and card p <= PedestransLimit
init ts == ts = mk_TSystem ({|->}, {|->}, {}, {}, {}, {}, {}, {})
end
```

The formal model performs some functionality in the overall system, in which the complete system depends on those functions and works properly. In our model, the traffic controller will change the signal using two functions, namely, vehicle and pedestrians that alter the signal time according to the current traffic situation.

```
unctions
ChangevehiclesSignal: (map route to TSignals) * route * TSignals -> (map route to TSignals)
ChangevehiclesSignal (tsignals, r, colour) == tsignals ++ {r |-> colour};
ChangePedestriansSignal: (map route to PSignals) * route * PSignals -> (map route to PSignals)
ChangePedestriansSignal (psignals, r, colour) == psignals ++ {r |-> colour};
```

The formal model performs various operations on this module; these operations include add a camera, add a sensor, remove camera, remove sensor, update traffic situations, dynamic time, pedestrians detection, and primary vehicle detection. The first operation is adding a camera, which has two inputs, namely, the first is a camera that is to be fitted on traffic signal intersection and the other is the camera location where the camera is to be fixed. Our precondition is to check that our camera is already used or not; if the camera is not being used, then it will map a camera with the new location.

The second operation is adding a sensor that has two inputs that sense the information on roads at a particular location and sends it to the traffic controller. The pre-condition is checking whether the sensor is already used or not. Thus, if the sensor is new, then the sensor is mapped to the location on road.

```
Addsensor (sensorin: Sensors, locin: Location)
ext wr location: map Sensors to Location
pre sensorin not in set dom location
post location = location munion{sensorin|->locin};
Addcamera (camerain: Cameras, clocin: CLocation)
ext wr clocation: map Cameras to CLocation
pre camerain not in set dom clocation
post clocation = clocation munion {camerain |-> clocin};
removesensor (sensorin: Sensors)
ext wr location: map Sensors to Location
pre sensorin in set dom location
post location = {sensorin} <: location;
removecamera (camerain: Cameras)
ext wr clocation: map Cameras to CLocation
pre camerain in set dom clocation post clocation = {camerain} <: clocation;
```

The third operation is to update the traffic situation that has one input of primary vehicles and give us the current road situation in quote type. The main purpose of this operation is to provide the best route in any situation. In this operation, the condition is to check three different limits and select a path from congestion, smooth, and traffic stopped.

```
Updatetrafficsituation (pvehicles: int)
ext wr currenttraffic: set of trafficSitutions
pre true
post ((pvehicles >= 0 or pvehicles <= 50) and currenttraffic ={<Congestion>}) or ((pvehicles >= 41
or pvehicles <= 30) and currenttraffic = {<Smooth>}) or
((pvehicles >=100) and currenttraffic ={<Stopped>});
```

The dynamic time operation has a single input that signals duration and returns the current traffic in quote type. The current traffic situation can be in different ways, such as smooth and congestion on roads. If the traffic situation is running smoothly, then the signal duration is decided accordingly, and if the current situation is congested, then the signal duration will be set accordingly.

```
Dynamictime (signalduration: int)
ext wr currenttraffic: set of trafficSitutions
Pre true
Post if currenttraffic = {<Smooth>}
Then signalduration = 10
elseif currenttraffic = {<Congestion>}
then signalduration = 20
else signalduration = 15;
```

The pedestrian detection operation has multiple inputs, including ptime, psignals, vtime, and tsignals. This operation returns the current situation for the primary vehicles and pedestrians in quote type. The current traffic type set is changed when the pedestrian's detection occurs or the primary vehicle's detection. If the traffic detector is detecting pedestrians, then the system does not give primary vehicles the permission to cross the lane; and if the system is detecting that the primary vehicles are more than the pedestrians, then primary vehicles are given the permission to cross the lane and display their signal light. Lastly, the signal duration will set accordingly.

```
Pedestriansdetection (ptime: int, psignals: PSignals, vtime: int, tsignals: TSignals)
ext wr traffictype: set of Traffictype
pre true
post if traffictype = {<Pedestrians>}
then psignals = <Green> and ptime = 10 and tsignals = <Red> and vtime = 0
elseif traffictype = {<Pedestrians>}
then psignals = <Red> and ptime= 0 and tsignals = <Green> and vtime = 20
```

The primary vehicle detection operation has multiple inputs, including vtime, tsignals, ptime, and psignals. This operation also returns the current situation for the primary vehicles and pedestrians in quote type. The current set of traffic type will change when the primary vehicle or the pedestrian is detected. If the traffic detector is detecting the primary vehicles more than pedestrians, then the primary vehicles are given the permission to cross the lane and set their traffic signal. At this time, pedestrians are not allowed to cross the lane. In case pedestrians are detected, normal vehicles cannot cross the lane and display their signal. Signal duration will be set accordingly.

```
primaryvehiclesdetection (vtime: int, tsignals: TSignals, ptime: int, psignals: PSignals)
ext wr traffictype: set of Traffictype
pre true
post if traffictype = {<PVehicles>}
then tsignals = <Green> and vtime = 20 and psignals = <Red> and ptime = 0
elseif traffictype = {<PVehicles>} and psignals = <Green> and ptime = 20
then tsignals = <Red> and vtime= 0
else true;
```

### 7.2. Emergency Module

Emergency module is another component of our traffic signaling system. To sense or identify real-time traffic situations, we have deployed sensors and cameras on traffic signals. However, what sensors and cameras will sense or monitor the emergency vehicles? This is a worldwide issue in many large countries or even large cities that tried to use several devices to find out the mechanism of emergency vehicles. We are providing different types of solutions for various emergency vehicles. A mechanism provides priority for the emergency vehicles on every lane, which is detected by the cameras and sensors, and then the system controller takes actions against the received information.

In the formal specification of the traffic system, one part has types in which variables with their quote types are declared. This model used various variables, such as traffic signals, emergency vehicle type, traffic situation, to obtain information by the sensor, which is either quote types or token types.

```
types
Id = token;
Path1 = token;
Path2 = token;
ESignals = <Red>|<Green>;
Signals = <Green>|<Red>;
Emergencylocation= token;
Emergencytype = <Fire>|<Ambulance>|<Accident>|<Police>;
Emergency = token;
```

```
Location = token;
CamLocation = token;
Sensors = token;
Name = token;
Cameras = token;
TrafficSitutions = <Stopped> | <Congestion> | <Smooth>;
Sensorinformation = token;
Getinformationbysensor = <Emergencytype> | <Emergencylocation>;
```

The most important composite objects are conflict and vehicles. Conflict object has an invariant for opposite routes that are not equal for same signals and equal time at the same time because this condition prevents the chances of accident. The emergency vehicle object has a vehicle ID and a set of emergency types and locations. This is a composite object of emergency vehicles that have all emergency information in the VDM-SL, and it is called the composite data type. It has an invariant that verifies its correct initialization and its integrity. Its invariant checks if an emergency vehicle is detected: if yes, then it must be located. If verified to be an emergency vehicle, then it must either be a vehicle, an ambulance, police, or an accident vehicle.

```
Conflict::P1: Path1
P2: Path2
inv mk_Conflict (P1, P2) == P1 <> P2;
Evehicles: evid: Id
etype: set of Emergencytype
eloc: set of Emergencylocation
inv mk_Evehicles (-, el, et) == (card(el) = card(et)) and (({<Fire>} subset el\{<Ambulance>} or
{<Ambulance>} subset el\{<Fire>}) or {<Accident>} subset el\{<Police>} or {<Police>} subset
el\{<Accident>} and {<Fire>, <Ambulance>, <Accident>, <Police>} inter el <> {});
```

The important part of the formal specification is called state. The state explains the behavior of the system, and it gives different values to variables. Different variables are defined with types in the state portion. The state has an invariant and initialization. Invariant is the condition that must be applied successfully before the declaration of objects, and initialization is the state that defined the values of all variables. In-state, loc is used for locations that are taken from sensors. The system that maintains sensors and cameras also have a current location variable if any emergency is currently going on.

```
state Emergencysystem of
sensors: set of Sensors
cameras: set of Cameras
emergencyvehicles: set of Emergencytype
loc: map Sensors to Location
camloc: map Cameras toCamLocation
reallocation: set of Location
allEmergencies: map Id to Emergencytype
getinformationbysensor: map Sensors to Getinformationbysensor
totalemergency: set of Emergency
init mk_Emergencsystem (-, -, ev, l, cl, rl, -, ae, te) == ev = {} and l= { | ->} and cl = { | ->} and
ae= { | ->} and rl= {} and te = {}
end
```

In our model, the change emergency signal function will perform the changing time of signals according to the current traffic situation.

```
functions
ChangeEmergencySignal: (map route to ESignals) * route * ESignals -> (map route to ESignals)
ChangeEmergencySignal(esignals, r, colour) == esignals ++ {r | -> colour};
```

Several operations that provide services to drivers when an emergency vehicle operates are included in the emergency facilities.

The formal model performs various operations on traffic systems; these operations include set sensors on road, set the camera on signal, emergency lane, new emergency, emergency vehicle, and obtain info for an emergency, new emergency occurrences, obtain specific emergency, and total emergency vehicles.

The first operation is adding a sensor, which has two inputs that sense the information on roads at a particular location and sends it to the traffic controller. The pre-condition is to check whether the sensor is already used or not. Hence, if the sensor is new, then the sensor is mapped to the location on the road. The second operation is adding a camera, which has two inputs; the first is a camera that needs to be fitted on the traffic signal intersection, and the other is the camera location where the camera must be fixed. Our precondition is whether the camera is already used or new. Hence, if the camera is not already used, then that camera is mapped to the specific location.

```
Setsensorsonroad (sensorIn: Sensors, locationin: Location)
ext wr loc: map Sensors to Location
wr sensors: set of Sensors
pre sensorIn not in set dom loc
post loc = loc munion {sensorIn |-> locationin};
setcameraonsignal (cameraIn: Cameras, locationin: CamLocation)
ext wr camloc: map Cameras to CamLocation
wr cameras: set of Cameras
pre cameraIn not in set dom camloc
post camloc = camloc munion {cameraIn |-> locationin};
```

The third operation is to find the emergency occurrence lane, which has three inputs. One is the camera for detecting the emergency vehicles, providing the type of emergency, and giving us the traffic signal in quote type. The main purpose of this operation is to find any types of emergencies and to enable the lane where the emergency occurs to change the signal into green, and the other will turn red so that the emergency vehicle that will cross will be given priority.

```
Emergencylane (cameraIn: Cameras, etype: Emergencytype, signals: Signals)
ext wr allEmergencies: map Id to Emergencytype
rd cameras: set of Cameras
pre cameraIn in set cameras
post if etype = <Fire>
then signals = <Green>
else signals = <Red>;
```

The next operation is related to an emergency if there is an emergency that occurs on the road, so this system is dealing with any emergency that returns true or false. If the sensor detects any ambulance, fire brigade, police, and accident, then the system will produce the emergency signal to alert other vehicles.

```
Emergencevehicle () emergencydetection: bool
ext wr evehicles: set of EVehicles
pre true
post if evehicles = {<Ambulance>}
then emergencydetection = true
else emergencydetection = false;
```

In a new emergency operation, it will find a new emergency and needs some parameters that are added to our system of all emergencies. To find a new emergency service, it needs ID, location for an emergency, and type of emergency.

```
Newemergency (evid: Id, eloc: Emergencylocation, etype: Emergencytype)
ext wr allEmergencies: map Id to Emergencytype
pre (evid in set dom allEmergencies)
post allEmergencies = allEmergencies munion {evid |-> mk_Evehicles (evid, {etype}, {eloc})};
```

Obtaining information for emergency by sensor operation provides us information about the emergency location through sensor. It will provide us information of all activities on roads so that information can be given to the system if any emergency occurs. This operation has a pre-condition that proves that the information of the specific accident is not already in the system database, and if it validates the condition, truly, then post condition will be executed, and the current information of the emergency will be included in our system.

```
Getinfoforemergency (getinfo: Getinformationbysensor, sensorIn: Sensors)
ext wr Getinformationbysensor: map Sensors to Getinformationbysensor
pre(getinfo not in set rng Getinformationbysensor) and (sensorIn in set dom loc)
post Getinformationbysensor= Getinformationbysensor munion {getinfo |-> sensorIn};
```

New emergency occurrences operation creates a new emergency; in the list of emergencies, some information about a new emergency is needed to be added to our system. It needs emergency ID, emergency type, and location of the emergency.

```
NewEmergencyoccurances (eid: Id, etype: Emergencytype, eloc: Emergencylocation)
ext wr allEmergencies: map Id to Emergencytype
pre (eid in set dom allEmergencies)
post allEmergencies = allEmergencies munion {eid |-> mk_Evehicles (eid, {etype}, {eloc})};
```

Obtain specific emergency operation will store all emergency services from the system. This operation determines the desired emergency situations from the list and provides it an ID of a specific emergency. This operation is mapped with ID in the list and obtains those elements and returns a specific emergency.

```
Getspecificemergency (eid: Id)
ext rd allEmergencies: map Id to Emergencytype
pre true
post allEmergencies = {eid} <: allEmergencies;
```

Total emergency vehicles operation provides the overall number of emergencies. This operation acts like a central information system that manages how many emergency vehicles are registered in the system. In this operation, we determine the number of vehicles, and we use the operator of cardinality for emergency vehicle sets. This operator shows the total number of emergency vehicles that belong to a set.

```
Totalemergencyvehicles () total: int
ext rd totalemergency: set of Emergency
pre true
post total = card totalemergency;
```

### 7.3. Shortest Path Module

In the proposed model, various objects, such as marts, hospitals, and schools, are included. They have certain identifiers, names, and locations.

```
types
Id = token; Name = token;
Route1 = token; Route2 = token;
Mart:: mid: Id
mname: Name
mloc: int
mtime: Time;
Hospital:: hid: Id
hname: Name
hloc: int
htime: Time;
School::sid: Id
sname: Name
sloc: int
```

```
stime: Time;
Time:: hour: nat
min: nat;
RouteSignals:: route1: Route1
route2: Route2
inv mk_RouteSignals (route1, route2) == route1<>route2;
```

In a smart city, roads and traffic signals always exist. The field rid means that each road has a unique identifier. The field traffic signal defines whether there are traffic signals on every traffic signalized intersection, such as green, yellow, and red. All traffic signal cameras have been deployed on every lane to monitor the real-time traffic situation. On every lane, sensors are deployed to sense the traffic and send information to the controller.

```
Green = token;
Yellow = token;
Red = token;
Roadintersection:: rid: Id
trafficsignal: Trafficsignal
sensor: Sensor;
Trafficsignal:: green: Green
yellow: Yellow
red: Red
cameras: Cameras;
Cameras:: cid: Id
cloc: int
detect: set of Vehicles;
Sensor:: sid: Id
sloc: int
detect: set of Vehicles;
```

All objects are assumed as nodes that ensure identifiers. The object links are described to define the connectivity of two objects in our network. The field roadintersections indicates that a connection lies in a set of roads. The field gedge shows the connectivity of two objects by an edge. A gedge contains two nodes in the network. In an invariant, a gedge consisted of two nodes in our network.

```
Network:: mart: Mart
hospital: Hospital
school: School
roadintersection: Roadintersection.
GNode:: nid: Id
network: Network;
GEdge:: E1: GNode
E2: GNode;
Links:: roadintersections: set of Roadintersection
gedge: GEdge
inv mk_Links (roadintersections, gedge) == forall ri1, ri2 in set roadintersections
& ri1 = gedge.E1.network.roadintersection and ri2 = gedge.E2.network.roadintersection;
```

The topology of this network is defined by graphconnection that contains the set of nodes specified, such as gnodes, and a set of gedges are shown as links. In this invariant, two nodes in our network are connected through a gedge.

```
GraphConnection:: gnodes: set of GNode
links: set of Links
inv mk_GraphConnection (gnodes, links) == forall n1, n2 in set gnodes & exists tlink
in set links & n1.network.roadintersection = tlink.gedge.E1.network.roadintersection and
n2.network.roadintersection = tlink.gedge.E2.network.roadintersection;
```

The complete network of the smart city is described as a traffic model that is graph-connection and vehicles.

```
VLicence = token;
Vehicles:: vid: Id
vlicence: VLicence;
TrafficModel:: graphconnection: GraphConnection
vehicles: set of Vehicles
sensor: set of Sensor
cameras: set of Cameras;
```

The most important part of VDM-SL is the state that is specified as TrafficSignalingSystem, which includes some attributes, as defined above in the type of objects. The initialization is used to initialize all objects.

```
state TrafficSignalingSystem of
trafficmodel: [TrafficModel]
vehicles: set of Vehicles
sensor: set of Sensor
cameras: set of Cameras
mart: set of Mart
hospital: set of Hospital
school: set of School
init tss == tss = mk_TrafficSignalingSystem (nil, {}, {}, {}, {}, {}, {})
end
```

The formal specification of some major operations is described below. To give permission for the vehicle to enter in a smart system, the operation Givepermissiontoentervehicle, which takes a vehicle and vehicle license as input, is defined. In the external clause, the system is read. It also changes the set of vehicles as the new vehicle becomes part of our system. In the pre-condition, the vehicle should have identifiers. Then, the vehicle can be included. Such a phenomenon is described in the post section in terms of the union operation of the portion of vehicles.

```
Givepermissiontoentervehicle (vehicle: Vehicles, vlicence: VLicence)
ext rd trafficmodel: [TrafficModel]
wr vehicles: set of Vehicles
pre vehicle not in set trafficmodel.vehicles and vehicle.vlicence = vlicence
post vehicles = vehicles~ union {vehicle};
```

Similarly, the vehicles are given permission to leave if they belong to a smart city, and they must have a vehicle license. In the post portion, vehicles are removed in a set of vehicles from the smart city.

```
Givepermissiontoleavevehicle(vehicle: Vehicles, vlicence: VLicence)
ext rd trafficmodel: [TrafficModel]
wr vehicles: set of Vehicles
pre vehicle not in set trafficmodel.vehicles and vehicle.vlicence = vlicence
post vehicles = vehicles~\{vehicle};
```

The operation Select low rush traffic signal intersection has low rush traffic; this operation is used to experience less traffic signal intersection. This operation has no true pre-condition. In our post-condition, all traffic signalized intersections from all nodes have few numbers of vehicles. This operation is returned as the low rush signal intersection used as output.

```
Selectlowrushsignalintersection () lowrushlane: Roadintersection
ext rd trafficmodel: [TrafficModel]
pre true
post forall n1 in set trafficmodel.graphconnection.gnodes & exists n2 in set
trafficmodel.graphconnection.gnodes &
n1.network.roadintersection = n2.network.roadintersection and card
n2.network.roadintersection.trafficsignal.cameras.detect < card
n1.network.roadintersection.trafficsignal.cameras.detect and
lowrushlane = n2.network.roadintersection;
```

The measurement of the shortest path based on time is specified. It takes less time, start travel, and destination nodes as inputs and produces the sequence of roadintersection as the output in the shortest path. In our post-condition, the first node in all node sequences of the graphconnection correspond to our start node/signal intersection. Then, all controllers of signalized intersections are giving information to the drivers so that they save their time and fuel. If the time of mart, school, and hospital, then that traffic signal intersection is not involved in the path and the hospital traffic signal may always be ignored in the route. Finally, the shortest path always depends on the return time.

```
Shortestpathbasedontime (lesstime: Time, starttravel: GNode, destinationnode: GNode)
lessrushpath: seq of Roadintersection
ext rd trafficmodel: [TrafficModel]
rd mart: set of Mart
rd hospital: set of Hospital
rd school: set of School
pre true
post forall grnode: seq of GNode & elems grnode =trafficmodel.graphconnetion.gnodes and forall
a in set inds grnode & starttravel = grnode (1) and lessrushpath () =
grnode(a).network.roadintersection and exists mat in set mart &
lesstime = mat.mtime => grnode(a).network.roadintersection.rid <> mat.mid
or exists scl in set school & lesstime = scl.stime => grnode(a).network.roadintersection.rid <>
scl.sid or exists hos in set hospital & lesstime= hos.htime =>
grnode(a).network.roadintersection.rid <> hos.hid and destinationnode = grnode(len grnode)
and grnode(a).network.roadintersection in set elems lessrushpath;
```

The shortest route depends on the distance that is measured in the shortest path based on distance. The start and destination nodes are taken as inputs, and the shortest route as output. In the post-condition, all the node sequences belong to graph connection, and the first start node is the start node. All the sequence nodes that have few rushes on traffic signalized intersection are returned. Lastly, the shortest route depends on the time that is provided as the output.

```
Shortestpathbasedondistance(starttravel: GNode, destinationnode: GNode) lessrushpath: seq of
Roadintersection
ext rd trafficmodel: [TrafficModel]
pre true
post forall grnode: seq of GNode & elems grnode = trafficmodel.graphconnection.gnodes and
forall
a in set inds grnode & starttravel = grnode(1) and lessrushpath() =
grnode(a).network.roadintersection
and destinationnode = grnode(len grnode) and grnode(a).network.roadintersection in set elems
lessrushpath;
```

The operations for searching a mart, hospital, and school are indicated to take their names as input and to return the location of all objects.

```
Searchmart (mname: Name) mloc: int
ext rd trafficmodel: [TrafficModel]
pre true
post exists mt in set trafficmodel.graphconnection.gnodes & mt.network.mart.mname= mname
and mloc = mt.network.mart.mloc;
Searchhospital (hname: Name) hloc: int
ext rd trafficmodel: [TrafficModel]
pre true
post exists hos in set trafficmodel.graphconnection.gnodes &
hos.network.hospital.hname = hname and hloc = hos.network.hospital.hloc;
Searchschool (sname: Name) sloc: int
ext rd trafficmodel: [TrafficModel]
pre true
post exists scl in set trafficmodel.graphconnection.gnodes & scl.network.school.sname = sname
and sloc = scl.network.school.sloc;
```

## 8. The Modeling Components and Properties of System

Here, we summarized the major modeling components and the properties of the system.

- The wireless sensors, cameras, primary vehicles, emergency vehicles, pedestrians, traffic, emergency, and pedestrians' signals are added to keep the updates in the real-time traffic situation and to provide responsive time to control the traffic environment in efficient manners.
- These are modeled using variables, types, composite types, quote types, invariants, initialization, states, functions, and operations.

Three modules are provided with their components that are used in formal specifications as following:

➢ Module 1: Responsive time module

Traffic signals will be changed with respect to emergency vehicles, pedestrians, and primary vehicles at the time of detection.

I. State Invariants

- The state has an invariant and initialization: invariant is the condition that must be applied successfully before the declaration of objects and initialization in the state that defined the values of all variables. Then the traffic system will check the primary vehicles that must be equal to or less than the fixed limit on one lane.

II. Functions

- In our model, the traffic controller will change the signal using two functions, namely, vehicle and pedestrians that alter the signal time according to the current traffic situation.

III. Operations

- The formal model performs various operations on traffic systems, as follows: add a camera, add a sensor, update traffic situations, dynamic time, and emergency vehicle. The description of each operation is provided in the first subsection of Formal specifications.

➢ Module 2: Emergency Services Module

There is a mechanism that will provide priority for the emergency vehicles on every lane that is identified by the cameras and sensors, and then the system controller takes actions against received information. This module provided emergency services which take place on traffic signals.

I. State Invariants

- The state has an invariant and initialization: invariant is the condition that must be applied successfully before the declaration of objects and initialization in the state that defined the values of all variables.
- In-state, loc is used for locations that are taking from sensors. The system maintains sensors, and cameras also have a current location variable if there is any emergency currently happening

II. Functions

- In our model, the change emergency signal function will perform the changing time of signals according to the current traffic situation.

III. Operations

- The formal model performs various operations on the traffic system: set sensors on road, set the camera on signal, emergency lane, new emergency, emergency vehicle, obtain info for an emergency, new emergency occurrences, obtain specific emergency, and total emergency vehicles. The working of each operation is given in the second subsection of Formal specifications.

➢ Module 3: Optimal Path Module

In the proposed model, various objects are included, such as mart, hospitals, and school. Drivers can choose the best way to identify the shortest path to reach the mart, hospital, and school. They have certain identifiers, names, and locations.

I.  Operations
   - The formal model performs various operations to find optimal paths across the smart city: give permission to enter vehicle, give permission to leave vehicle, select low rush signal, shortest path based on time, shortest path based on distance, search mart, search hospital, search school. The explanations of every operation are provided in the third subsection of Formal specifications.

II.  Properties of system
   - ■ Traffic signals will be changed with respect to emergency vehicles, pedestrians, and primary vehicles at the time of detection which are specified in module 1, 2, 3.
   - ■ When the emergency vehicle is detected, the controller will alert each lane to stop the flow of their vehicles and display the red signals on all lanes to continue the flow of the emergency vehicle, which is specified in module 2.
   - ■ When the need of any emergency services arises, a controller will identify the old emergency, new emergency, total emergencies, and information about specific emergency, which is specified in module 2.
   - ■ The number of waiting vehicles will be counted at each lane, and then a responsive traffic signaling scheme will be set that based on the analysis of controllers which are specified in module 1.
   - ■ In every vehicle, the driver will see two or more ways to reach any destination and find the suitable path to save time and fuel which are specified in module 3.
   - ■ When the controller detects any pedestrian on the footpath near the zebra crossing, all the signals will be turned red to inform all vehicles to stop, thereby enabling the pedestrians to cross the road safely which are specified in module 1.

## 9. Formal Analysis with VDM-SL Toolbox Facilities

The formal specifications of both static and dynamic modules are analyzed using the available techniques in the VDM-SL toolbox. The major techniques are Syntax, Type checker, C++ code generator, pretty printer and integrity properties analyzed it. Through checking all the properties, no errors are reported, which guarantees the accuracy of the developed model.

The VDM tool is a toolbox used for the development of system-oriented formal specifications in VDM-SL and an object-oriented extension of the VDM-SL. VDM Tools supports lots of several properties, from a basic syntax checker to the creation of systems from Java code.

We have formally analyzed and verified all the modules, including responsive time, mechanism for emergency services, and shortest path module. The formal specification is based on basic mathematical notations like mapping, set, sequence, predicate logic, and relations. The proof of accurate formal specification is provided using all facilities of VDM-SL toolbox.

In this article, the VDM-SL toolbox verifies the properties that are provided by the tool. The tool analyzes many facilities (like Syntax checker, Type checker, C++, Pretty printer). The formal specifications have been written in a Word document with the extension of Rich Text Format. The analysis of formal specifications has performed in VDM-SL toolbox. Our formal model contains a standard template as shown in output diagram. First, types, composite types, are defined. Second is defined the state. Third is declared functions. Forth is performed operations on state and variables using functions.

There is not a tool that exists which completely guarantees the whole correctness of any system model. It helps in identification of the possible errors at initial stages of any software development.

The developed system is formally analyzed and verified using VDM-SL toolbox. Verification and validation are two major pillars for the development of any system. Verification guarantees that a developed system at that stage fulfills the requirements calculated in a previous stage, while in the validation process, it is confirmed whether the generated system satisfies the user requirements. Dynamic time, mechanism for primary vehicles and pedestrians, emergency services, and optimal path are verified and validated using the VDM-SL toolbox properties. The formal specification analysis is done by verifying the syntax, type errors and integrity checkers.

The modules of Formal Analysis of Responsive Time, Emergency services and shortest path with their properties have been verified using the VDM-SL toolbox.

In Tables 1–3, we have formally verified all operations of three modules in our formal model through the VDM-SL toolbox features. The analysis of operations is accurately provided with verifying of syntax free, type free, c++, integrity checker and pretty printer. All tables provide proof of the accuracy of each operation in which the system is developed.

**Table 1.** Analysis of Responsive Time.

| Operations | Syntax Checker | Type Check | Integrity Check | C++ | Pretty Print |
|---|---|---|---|---|---|
| Add camera | Yes | yes | Yes | yes | Yes |
| Remove camera | Yes | Yes | Yes | Yes | Yes |
| Add sensor | Yes | Yes | Yes | Yes | Yes |
| Remove sensor | Yes | Yes | Yes | Yes | Yes |
| Update traffic situation | Yes | Yes | Yes | Yes | Yes |
| Dynamic time | Yes | Yes | Yes | Yes | Yes |
| Pedestrians' detection | Yes | Yes | Yes | Yes | Yes |
| Primary vehicle detection | Yes | Yes | Yes | Yes | Yes |

**Table 2.** Analysis of Emergency Vehicle Services.

| Operations | Syntax Checker | Type Check | Integrity Check | C++ | Pretty Print |
|---|---|---|---|---|---|
| Set sensor on road | Yes | yes | Yes | yes | Yes |
| Set camera on road | Yes | Yes | Yes | Yes | Yes |
| Emergency lane | Yes | Yes | Yes | Yes | Yes |
| Emergency vehicle | Yes | Yes | Yes | Yes | Yes |
| New emergency | Yes | Yes | Yes | Yes | Yes |
| Get info for emergency | Yes | Yes | Yes | Yes | Yes |
| New emergency occurrences | Yes | Yes | yes | Yes | Yes |
| Get specific emergency | Yes | Yes | Yes | Yes | Yes |
| Total emergency vehicles | Yes | yes | yes | yes | Yes |

**Table 3.** Analysis of Shortest Path.

| Operations | Syntax Checker | Type Check | Integrity Check | C++ | Pretty Print |
|---|---|---|---|---|---|
| Give permission to enter vehicle | Yes | yes | Yes | yes | Yes |
| Give permission to leave vehicle | Yes | Yes | Yes | Yes | Yes |
| Select low rush signal | Yes | Yes | Yes | Yes | Yes |
| Shortest path based on time | Yes | Yes | Yes | Yes | Yes |
| Shortest path based on distance | Yes | Yes | Yes | Yes | Yes |
| Search mart | Yes | Yes | Yes | Yes | Yes |
| Search hospital | Yes | Yes | Yes | Yes | Yes |
| Search school | Yes | Yes | Yes | Yes | Yes |

## 10. Conclusions

This paper presents the responsive time of primary vehicles at every signal to save time and fuel. Emergency vehicle services enable drivers to provide a path or a specific lane and save the lives of injured people. The facility of the shortest path to provide the information about congestion or fewer rush roads depends on time and distance; in this way, drivers can select a path according to their needs. The major results achieved are as follows: identified the real-time traffic signalized intersection information, to inform the road drivers about the emergency vehicles and shortest path, and to provide the responsive time at every signal to drive smoothly and reduce the waiting time. Real-time traffic is examined using sensors and cameras on traffic signal intersections to inform drivers about the facilities and to inform the local administrator to update traffic information about vehicles and traffic signals. In our model, the emergency module works with emergency services to overcome different incidents, such as fire and accidents. Traffic lights are organized to ensure the smooth flow of vehicles. Several sensors with cameras are configured and distributed on different traffic signals, schools, banks, and marts for continuously monitoring and making effective responsive traffic systems in real-time on roads Moreover, the vehicle drivers are reported to select alternative routes to find their destination in terms of saving time and fuel. In our proposed model, formal method-based VDM-SL approaches are used to develop the formal specification of this system. The components of the developed model are verified, validated, and analyzed by VDM-SL toolbox properties. Furthermore, this system will be improved to facilitate smart services in traffic signaling systems in any country.

## 11. Future Work

In the future, we will address some aspects such as carbon footprint, volume of emissions, means of mitigation, etc. We will apply model checking on different WSAN-based components to enhance efficiency in traffic signals and verify the system properties in real-life.

**Author Contributions:** Conceptualizations, A.N. and N.A.Z.; Introduction, A.N. and N.A.Z.; Background, A.N., N.A.Z. and E.H.A.; Our Contributions, A.N., N.A.Z. and E.H.A.; Related work, A.N., N.A.Z. and E.H.A.; System Architecture, A.N. and E.H.A.; Introduction to Formal Modelling, A.N., E.H.A. and N.A.Z.; Formal Specification, A.N. and E.H.A.; The Modeling Components and Properties of System, E.H.A.; A.N. and N.A.Z.; Formal Analysis with VDM-SL Toolbox Facilities. All authors have read and agreed to the published version of the manuscript.

**Funding:** This work is supported by Taif University Researchers Supporting Project number (TURSP-2020/292) Taif University, Taif, Saudi Arabia.

**Institutional Review Board Statement:** Not applicable.

**Informed Consent Statement:** Not applicable.

**Data Availability Statement:** Not applicable.

**Conflicts of Interest:** The authors declare no conflict of interest.

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
