# Peer review of "Formal Modeling of Responsive Traffic Signaling System Using Graph Theory and VDM-SL"

_sustainability, doi:10.3390/su132111772_

Round 1
Reviewer 1 Report
Formal modeling of traffic signaling system is interesting, while the motivation should be further highlighted. Moreover, the theoretical aspect of this paper can be further improved.
Author Response
Thank you for giving us a valuable comments.
Reviewer 2 Report
1. I would like to understand whether the survey was used with actual data from the city of Vienna and whether the participating persons (303 surveyed) have given their express authorization to use their data in accordance with the General Data Protection Regulation (GDPR);
2. Adherence to Smart Cities or Sustainable Cities was not evidenced, considering that at no time does the article address aspects such as carbon footprint, volume of emissions, means of mitigation, among others. The article focuses on computational parameters (specification of requirements with the UML language, algorithms in graphs, Bayesian network, WSAN, among others) of the transit system.
3. In lines 319 and 320, there is the central point of the article that should be explored for submission to this journal. Sustainable traffic systems become interesting topics if linked to environmental issues in cities.
4. The article does not bring any innovation or statistics of implementations about sustainability processes in neighborhoods, cities or regions cited as sources of the study.
5. The article seems more like a proposal for the implementation and validation of a software than a scientific article, therefore, it does not meet the minimum requirements for publication in this journal.
Author Response
Thank you for your comment.

Reviewer 3 Report
Novelty:
The objective of the manuscript is to establish a robust traffic monitoring and signaling system. The proposed method improves signal efficiency using Vienna Development Method Specification Language (VDM-SL) formal method and graph theory. The total waiting time at each signal intersection is reduced, and efficiency is optimized by applying a responsive traffic strategy. The model is validated and analyzed using many facilities available in the VDM-SL toolbox.
Weakness:
1. The texts in many figures are not clear.
2. Section 5 (Formal Specifications) is not succinct. There are too many codes embedded in the manuscript, which should be simplified in the following submission.
3. The VDM-SL toolbox figures in Section 8 (Formal Analysis with Outputs) are not necessary. You should demonstrate your analysis with detailed descriptions.
4. The experiment results are not sufficient enough to backbone the paper’s claims. Please add details to illustrate why the experiment results support the paper's objectives.
Round 2
Reviewer 2 Report
The article presented is not adhering to the journal and must be resubmitted to MDPI's Sensors.
Reviewer 3 Report
I am overall satisfied with the revision.